# Do components of adult height predict body composition and cardiometabolic risk in a young adult South Asian Indian population? Findings from a hospital-based cohort study in Pune, India: Pune Children's Study

Kalyanaraman Kumaran [1,2] Suyog M Joshi,[3] Chiara Di Gravio,[4] Himangi Lubree,[5] Charudatta Joglekar,[6] Dattatray Bhat,[3] Arun Kinare,[7] Ashish Bavdekar,[8] Sheila Bhave,[8] Anand Pandit,[8] Clive Osmond,[1] Chittaranjan Yajnik [3] Caroline Fall[1]

For numbered affiliations see end of article.

**Correspondence to**
Dr Chittaranjan Yajnik;
csyajnik@gmail.com

## ABSTRACT

**Objectives** We investigated whether the relationship between components of height and cardiovascular disease (CVD) risk may be explained by body composition. We also examined relationships between parental heights and offspring CVD risk.

**Design** A cohort study using cross-sectional data.

**Setting** A secondary care hospital setting in Pune, India.

**Participants** We studied 357 young adults and their parents in the Pune Children's Study. Primary and secondary outcomes: we measured weight, total height, leg length, sitting height, plasma glucose, insulin and lipids, and blood pressure (BP). Total and regional lean and fat mass were measured by dual X-ray absorptiometry.

**Results** Leg length was inversely related, and sitting height was directly related to BMI. Total height and leg length were directly related to lean mass, while sitting height was directly related to both lean and fat mass. Leg length was inversely related to systolic BP and 120 min glucose, independent of lean and fat mass. Sitting height was directly related to systolic BP and triglycerides; these relationships were attenuated on adjustment for lean and fat mass. When examined simultaneously, greater leg length was protective and greater sitting height was associated with a more detrimental CVD risk profile.

**Conclusions** Shorter adult leg length and greater sitting height are associated with a more adverse CVD risk factor profile. The mechanisms need further study, but our findings suggest a role for lean and fat mass.

## INTRODUCTION

Short adult height is associated with an increased risk of cardiovascular disease (CVD) and its risk factors in both high-income and low-income and middle-income countries (LMICs),[1–4] although the association is less consistent in the latter.[5 6] The reasons for the

## Strength and limitations of this study

► One of few birth cohorts in a low-income and middle-income country setting with data on parental and offspring heights, and height components.
► Data available on body composition measurements using dual-energy X-ray absorptiometry.
► Glucose–insulin measurements available as part of oral glucose tolerance test.
► Participants born in one hospital in Pune, India.
► ~25% loss to follow-up between childhood and young adulthood.

association remain unknown. Intrauterine and infant growth are determinants of final height.[7 8] Impaired fetal and infant growth are also associated with CVD.[9] It is therefore possible that the association between height and CVD is related to early life programming. Short height may also reflect lower socioeconomic class[10] and deprivation throughout childhood. The relationship between height and CVD may have a genetic basis; height is partly genetically determined as are some CVD risk factors. Finally, 'reverse causality' (ie, common risk factors for osteoporosis and CVD, which may lead to height shrinkage) has been suggested as a possible reason for the relationship between height and CVD.[11]

The components of height, that is, leg length and sitting height, have shown independent and contrasting relationships with CVD risk factors. Longer leg length has been associated with lower CVD risk,[12–17] while greater sitting height has shown either no

association[12 13] or an association with higher risk.[15] Most of these studies have been carried out in high-income countries. However, the reasons for these differential associations remain speculative. It has been suggested that during intrauterine insults, blood flow to the brain is selectively preserved at the expense of the peripheral organs, including the limbs ('brain-sparing effect').[18] This may result in shorter legs. It has also been suggested that increases in leg length occur mainly during infancy and childhood reflecting better nutrition and environmental circumstances during early life.[18 19] These factors might explain the relationship between leg length and CVD risk. In contrast, sitting height increases mainly during the pubertal growth spurt when the trunk and vertebral bones continue to grow after the limbs have stopped growing.[19] Accelerated pubertal growth and earlier onset of menarche are associated with greater CVD risk.[20 21] Shorter leg length and greater sitting height have also been associated with greater Body Mass Index (BMI) and/or higher body fat per cent.[22 23] It is therefore possible that the association between height components and CVD is influenced by body composition. Direct measurements of lean and fat mass reflect body composition better than BMI alone.

The Pune Children's Study (PCS) was set up to prospectively study early-life antecedents of adult disease.[24] At 8 years of age, taller children had a more adverse CVD risk profile.[24] We have now investigated the relationship of final height and its components with CVD risk factors at 21 years of age (height components were not assessed at 8 years). We hypothesise that (1) shorter leg length and greater sitting height will be associated with a more adverse CVD risk profile; (2) body composition measurements (lean and fat mass) may explain the relationship between height components and CVD risk. To our knowledge, this has not been examined before. We also investigated CVD risk factors in relation to parental height and intergenerational change in height.

## METHODS

The PCS[17] is an urban cohort of 477 full-term singleton babies who were born in the KEM Hospital, Pune, during 1987–1989. During 2009–2011, the cohort members were invited for further studies. They were admitted to our unit the evening before the investigations and fasted overnight (~10 hours) after a standard dinner. Those who were pregnant at the time of invitation were seen ~6 months after delivery for investigations.

### Patient and public involvement

It was not possible to involve participants in the design or conduct of the research project. The participants consisted of a birth cohort of men and women born in the KEM Hospital and who had been followed up longitudinally since early childhood. Results of the investigations (where clinically relevant) have been reported back to participants and appropriate referrals arranged for those requiring clinical management. Summary results of the research findings have been shared with the cohort.

### Anthropometry and body composition

Weight was measured to the nearest 5 g using an electronic scale (Conweigh Electronic Instruments, Mumbai). Standing and sitting height were measured to the nearest 0.1 cm using a wall-mounted stadiometer. Leg length was computed by subtracting sitting height from standing height. The predicted height of the offspring (mid-parental height) was derived using a standard formula (boys: [{maternal height+paternal height}/2]+7 cm; girls: [{maternal height+paternal height}/2]−7 cm.[25] The intergenerational change in height between children and parents was calculated by subtracting mid-parental height from offspring adult height. Participants' total and regional body fat and lean mass were assessed using dual-energy X-ray absorptiometry (Lunar Prodigy, GE, USA).

### CVD risk factors

Blood pressure (BP) was measured using a digital monitor (UA 767PC; A & D Instruments Ltd, UK); the average of two readings made 5 min apart was used. Plasma lipids, glucose and insulin were measured on fasting venous blood. An oral glucose tolerance test was carried out according to the WHO protocol,[26] followed by further blood samples at 30 and 120 min for glucose and insulin.

The Standard of Living Index (SLI)[27] was used to assess socioeconomic status. It is a standardised questionnaire based on information about housing, amenities and possessions; higher scores indicate higher social class.

### Laboratory analyses

Plasma glucose, cholesterol, high-density lipoprotein (HDL) cholesterol and triglyceride concentrations were measured using enzymatic methods (Hitachi 902, Germany). Between-batch coefficients of variation for all these assays were <3% in the normal range. Plasma insulin was measured using a Delfia technique (Victor 2, Wallac, Finland); between-batch coefficients of variation were <6%. Homeostatic model assessment of insulin resistance (HOMA-IR) and homeostatic model assessment of beta-cell function (HOMA-B) were calculated using the online Oxford model.[28] Insulin secretion was measured as Insulinogenic Index (increment in plasma insulin/increment in plasma glucose at 30 min).[29] The Matsuda Index of insulin sensitivity was computed by k/sqrt(-fasting glucose×120 min glucose×fasting insulin×120 min insulin), where k=10 000.[30] Disposition Index was calculated as Insulinogenic Index×Matsuda Index, to reflect beta-cell function for concurrent insulin resistance as originally described by Bergman.[31]

### Definitions

Overweight was defined as BMIs of ≥25 and <30 kg/m$^2$, and obesity as a BMI of ≥30 kg/m$^2$.[32] Stunting was defined as a height Z-score of less than −2 below the Centers for Disease Control and Prevention (CDC) average at 20 years of age.[33] Hyperglycaemia was defined as either

impaired fasting glucose (IFG, fasting plasma glucose 5.6–6.9 mmol/L) or impaired glucose tolerance (IGT, 120 min plasma glucose 7.8–11.0 mmol/L) or diabetes mellitus (DM, fasting plasma glucose ≥7.0 mmol/L or 120 min plasma glucose ≥11.1 mmol/L).[34] Hypercholesterolaemia was defined as plasma total cholesterol of ≥5.1 mmol/L,[35] hypertriglyceridaemia as a plasma triglyceride concentration of ≥1.7 mmol/L and low HDL cholesterol as HDL cholesterol concentration of <1.03 mmol/L for men and <1.29 mmol/L for women.[36] Hypertension was defined as systolic BP of ≥130 mm Hg or diastolic BP of ≥85 mm Hg.[36]

## Statistical methods
Exposure variables (offspring height and its components, parental height and intergenerational change in height) were converted into within-sample Z-scores to enable comparison of effects. Associations with body composition were tested using age-adjusted and sex-adjusted partial correlations. Multiple linear regression was used to identify associations between height components and CVD risk factors, adjusting for age, sex and age–sex interaction (model 1). We further adjusted for fat mass, lean mass and their interactions with sex (model 2). The estimated regression coefficients (β) represent the change in outcome per SD change in exposure. We examined the simultaneous association of height components and derived contour plots for the outcome variables as a function of leg length and sitting height, adjusted for age and sex. We used the Mahalanobis distance to identify the ellipse that contained 95% of the data and to exclude unlikely combinations of leg length and sitting height.

To test possible selection bias, we compared body size, lipid and glucose–insulin variables between participants and non-participants using regression imputation. We developed an imputation model using multiple regression with variables significantly associated with each other at 8 years. We then applied this model to compare 21-year observed values for participants and imputed values for non-participants.

Statistical analyses were performed using SPSS V.21 and R V.3.2.3.

## RESULTS
Of 477 children who were studied at 8 years of age, 357 (75%) participated in the 21-year study (191 men). Non-participants had higher 8-year BMI (14.0 vs 13.6 kg/m², p=0.05), but similar 8-year height, and lipid, glucose and insulin concentrations compared with participants. Participants' 21-year BMI showed no differences from imputed values for non-participants (21.6 vs 21.9 kg/m², p>0.05).

## General characteristics
Both men and women had height CDC Z-scores below the average (table 1). They were taller than their parents (men by 5.7 cm and women by 4.1 cm) with lower rates of stunting (8.0% and 14.5% of men and women vs 26.8% and 28.5% of fathers and mothers, respectively). Compared with their fathers, sons had a greater leg length by 3.7 cm and sitting height by 2.5 cm. Compared with their mothers, daughters had a greater leg length by 2.5 cm and sitting height by 1.3 cm. Offspring height correlated positively with the height of both parents (men: r=0.43 and 0.55 for fathers and mothers, respectively; women: 0.50 and 0.48; p<0.001 for all). Heights of the parents were positively correlated (r=0.23, p<0.001). At 21 years, 18.5% of participants were overweight; 2.5% were obese; 4.8% were hypertensive; 5.6% had hypercholesterolaemia; 7.6% had hypertriglyceridaemia; and 69.3% had low HDL cholesterol concentrations. None were on treatment for hypertension or dyslipidaemia. Three participants known to have diabetes were on insulin therapy; a further 18.5% were found to be hyperglycaemic (11.2% IFG, 5.9% IGT and 1.4% type 2 DM).

## Relationships of height with birth weight, socioeconomic status, BMI and body composition
Total height, leg length and sitting height at 21 years were positively associated with birth weight (r=0.39, 0.34 and 0.33, respectively; p<0.001 for all). Higher SLI scores were associated with taller total height (r=0.11, p<0.05); correlations with leg length and sitting height were similar (0.09 and 0.10, p>0.05). Leg length and leg length to sitting height ratio were negatively associated with BMI (r=−0.18 and r=−0.30, respectively), while sitting height was positively associated (r=0.17, p<0.01 for all). Total height and leg length were positively associated with lean mass, while sitting height was positively associated with both lean and fat mass (table 2). Leg length to sitting height ratio was inversely related to fat mass and body fat per cent. We examined the relationship of height components with regional body composition measurements (arm lean and fat mass, leg lean and fat mass, and trunk lean and fat mass); the findings were similar to those seen with total lean and fat mass (online supplemental figure 1). Height and leg length were positively associated with regional lean mass measurements, while sitting height was positively associated with both regional lean and fat mass measurements. Leg length to sitting height ratio was negatively associated with regional fat mass measurements.

## Relationships of height and its components with CVD risk factors
Relationships between height components and CVD risk factors were similar in both sexes, and there were no interactions with sex; we therefore present a sex-adjusted combined analysis. Total height was negatively associated with 120 min glucose concentration and Matsuda Index (table 3). Leg length was negatively associated with diastolic BP, total cholesterol, fasting and 120 min glucose concentrations, and positively associated with Matsuda Index. Sitting height was positively associated with systolic BP and triglycerides. Leg length to sitting height ratio

**Table 1**  Characteristics of the cohort participants

| | Men (n=191) | Women (n=166) |
|---|---|---|
| **General characteristics** | | |
| Birth data | | |
| Birth weight (g) | 2824 (443) | 2721 (481) |
| 21-year data | | |
| Age (years) | 21.4 (0.4) | 21.4 (0.4) |
| Anthropometry | | |
| Height (cm) | 172.0 (6.6) | 156.8 (6.4) |
| CDC Z-score | −0.66 (0.92) | −0.99 (0.98) |
| Stunted, n (%) | 15 (8.0) | 24 (14.5) |
| Sitting height (cm) | 89.8 (3.2) | 82.5 (3.1) |
| Leg length (cm) | 82.2 (4.4) | 74.2 (4.3) |
| Leg length:sitting height ratio | 0.91 (0.04) | 0.89 (0.04) |
| Weight (kg) | 65.3 (13.1) | 52.3 (10.6) |
| BMI (kg/m$^2$) | 22.0 (4.0) | 21.2 (4.1) |
| Lean mass (kg) | 46.5 (5.8) | 30.2 (3.9) |
| Fat mass (kg) | 15.7 (9.1) | 19.1 (8.0) |
| Fat per cent (%) | 22.4 (9.6) | 35.2 (8.4) |
| Cardiometabolic risk factors | | |
| Systolic BP (mm Hg) | 115.3 (10.1) | 102.9 (9.1) |
| Diastolic BP (mm Hg) | 65.4 (8.4) | 63.6 (7.9) |
| Cholesterol (mmol/L) | 3.8 (0.76) | 3.75 (0.74) |
| Triglycerides (mmol/L)* | 0.89 (0.67 to 1.25) | 0.76 (0.62 to 1.03) |
| HDL cholesterol (mmol/L) | 0.97 (0.17) | 1.10 (0.22) |
| Fasting glucose (mmol/L) | 5.1 (0.6) | 5.0 (0.4) |
| 120 min glucose (mmol/L)* | 5.5 (4.8 to 6.5) | 5.7 (4.8 to 6.5) |
| Fasting insulin (pmol/L)* | 41.4 (26.1 to 64.2) | 48.0 (31.8 to 68.4) |
| HOMA-IR* | 0.9 (0.6 to 1.4) | 1.0 (0.7 to 1.5) |
| HOMA-B* | 85.1 (60.2 to 106.1) | 94.5 (72.4 to 122.5) |
| Insulinogenic Index* | 52.2 (37.8 to 78.2) | 58.1 (39.1 to 83.1) |
| Matsuda Index* | 6.0 (3.8 to 8.9) | 5.2 (3.5 to 8.0) |
| Disposition Index* | 330.4 (191.5 to 512.5) | 290.9 (193.0 to 520.3) |
| Parental characteristics | | |
| | Fathers (n=310) | Mothers (n=360) |
| Age (years) | 52.5 (4.2) | 46.2 (4.5) |
| Height (cm) | 165.9 (6.1) | 153.0 (5.7) |
| Sitting height (cm) | 87.3 (3.3) | 81.4 (3.09) |
| Leg length (cm) | 78.6 (4.0) | 71.6 (3.7) |
| CDC Z-score | −1.52 (0.84) | −1.58 (0.88) |
| Stunted n (%) | 82 (26.8) | 102 (28.5) |

Values are mean (SD) unless otherwise stated.
*Median (IQR).
BMI, Body Mass Index; BP, blood pressure; CDC, Centers for Disease Control and Prevention; HDL, high-density lipoprotein; HOMA-B, homeostatic model assessment of β-cell function; HOMA-IR, homeostatic model assessment of insulin resistance.

was negatively associated with systolic and diastolic BPs, total cholesterol, triglycerides, and fasting and 120 min glucose concentrations, and positively associated with Matsuda Index. After adjustment for lean and fat mass, the relationships of height and leg length with risk factors remained of similar strength except for Matsuda Index, and a negative association of leg length with systolic BP became apparent. The relationships of sitting height

**Table 2** Relationship of height and components of height to body composition

| | Lean mass (kg) | | | Fat mass (kg) | | | Body fat % | | |
|---|---|---|---|---|---|---|---|---|---|
| | β | 95% CI | P value | β | 95% CI | P value | B | 95% CI | P value |
| **Men** | | | | | | | | | |
| Total height (z) | 3.6 | 3.0 to 4.3 | <0.001 | 1.2 | −0.1 to 2.5 | 0.07 | 0.1 | −1.3 to 1.5 | 0.9 |
| Leg length (z) | 2.8 | 2.1 to 3.6 | <0.001 | 0.4 | −1.0 to 1.7 | 0.6 | −0.5 | −1.9 to 0.9 | 0.5 |
| Sitting height (z) | 3.6 | 2.9 to 4.3 | <0.001 | 2.1 | 0.8 to 3.4 | 0.001 | 1.0 | −0.4 to 2.4 | 0.1 |
| Ratio of leg length to sitting height (z) | 0.5 | −0.3 to 1.4 | 0.2 | −1.2 | −2.5 to 0.1 | 0.08 | −1.4 | −2.7 to 0.0 | 0.05 |
| Women | | | | | | | | | |
| Total height (z) | 2.0 | 1.5 to 2.5 | <0.001 | 1.1 | −0.1 to 2.3 | 0.08 | −0.2 | −1.5 to 1.1 | 0.8 |
| Leg length (z) | 1.4 | 0.8 to 1.9 | <0.001 | −0.6 | −1.8 to 0.7 | 0.4 | −1.6 | -2.9, to -0.3 | 0.01 |
| Sitting height (z) | 2.3 | 1.8 to 2.8 | <0.001 | 2.8 | 1.6 to 4.0 | <0.001 | 1.5 | 0.2 to 2.9 | 0.02 |
| Ratio of leg length to sitting height (z) | 0.0 | −0.6 to 0.6 | 0.9 | −2.9 | -4.0, to -1.7 | <0.001 | −3.3 | -4.5, to -2.1 | <0.001 |

Data derived using linear regression and all variables as continuous; All analyses were adjusted for age.

were significantly attenuated by body composition adjustment. The relationships of leg length to sitting height ratio were partially attenuated after adjustment for body composition. Regional body composition measurements (trunk fat and lean mass for sitting height, or leg fat and lean mass for leg length) did not attenuate these associations more than total fat and lean mass (data not shown). We then adjusted for all the regional body composition measurements instead of total lean and fat mass while examining the relationship between height components and cardiometabolic risk factors. There were no specific patterns with regional body composition measurements, and these models did not explain the relationships between height components and CVD risk factors any more than total body composition measurements. These relationships remained similar after adjusting for birth weight and SLI.

### Relationships of CVD risk factors with leg length and sitting height simultaneously

The associations of leg length and sitting height simultaneously with CVD risk factors are shown in table 3 and as contour plots in figure 1 and online supplemental figures 2 and 3 . Because of the opposing relationships of leg length and sitting height with CVD risk factors, these associations tended to be accentuated. In general, at any leg length, greater sitting height was associated with a more adverse profile; at any sitting height, greater leg length was associated with a more favourable profile. The healthiest profiles tended to be in people with long legs and short sitting height. However, there was variation in patterns. For lean mass, systolic BP, triglycerides, HDL cholesterol and fasting glucose (contour lines near-horizontal), the associations with sitting height were stronger than those with leg length (figure 1). For 120 min glucose and Matsuda Index (contour lines near-vertical), the associations were stronger with leg length. For BMI, fat mass, total cholesterol and diastolic BP (contour lines near-diagonal), the associations with leg length and sitting height, though in

opposite directions, were of similar strength. There were no relationships with HOMA-IR, HOMA-B, Insulinogenic Index or Disposition Index.

### Relationships with parental height and intergenerational change in height

There were no relationships between height of either parent or mid-parental height with offspring CVD risk factors (table 4). A greater increase in leg length between generations was associated with lower BMI, fasting insulin, and HOMA-IR while a greater increase in sitting height was associated with higher BMI. There were no interactions between parental and offspring heights on offspring CVD risk factors.

### DISCUSSION

There was an intergenerational increase in total height, leg length and sitting height in this young adult Indian urban cohort. Greater total adult height and leg length were significantly associated with lower BMI, greater lean body mass and lower CVD risk factors, while a greater sitting height was significantly associated with higher BMI, lean and fat mass and higher CVD risk factors. These associations were accentuated when leg length and sitting height were examined simultaneously, and leg length to sitting height ratio was strongly negatively related to CVD risk factors. The relationships with sitting height were significantly attenuated by adjusting for lean and fat mass. Parental height was unrelated to offspring CVD risk factors; however, a greater intergenerational increase in leg length was associated with lower insulin and insulin resistance.

The associations of shorter leg length with higher CVD risk factors in our study are consistent with findings from high-income countries which have shown inverse associations of leg length with BP, glucose intolerance and dyslipidaemia.[11–17] The associations with sitting height are less consistent, with some studies showing positive

**Table 3** Relationship of components of height to cardiovascular disease risk factors

| Outcome | Model | Height (z) | | | Leg length (z) | | | Sitting height (z) | | | Leg length to sitting height (z) | | |
|---|---|---|---|---|---|---|---|---|---|---|---|---|---|
| | | B | 95% CI | P | B | 95% CI | P | B | 95% CI | P | B | 95% CI | P |
| Systolic BP (mm Hg) | | | | | | | | | | | | | |
| | 1 | 0.36 | (−0.66 to 1.38) | 0.49 | −0.77 | (−1.79 to 0.25) | 0.14 | 1.79 | (0.79 to 2.80) | 0.001 | −2.18 | (−3.18 to −1.19) | <0.001 |
| | 2 | −1.17 | (−2.34 to 0.00) | 0.05 | −1.74 | (−2.82 to −0.67) | <0.001 | 0.33 | (−0.87 to 1.54) | 0.58 | −1.74 | (−2.73 to −0.76) | <0.001 |
| Diastolic BP (mm Hg) | | | | | | | | | | | | | |
| | 1 | −0.36 | (−1.20 to 0.49) | 0.41 | −1.11 | (−1.94 to −0.27) | 0.01 | 0.77 | (−0.07 to 1.62) | 0.07 | −1.82 | (−2.65 to −1.00) | <0.001 |
| | 2 | −0.50 | (−1.46 to 0.45) | 0.30 | −0.93 | (−1.81 to −0.05) | 0.04 | 0.50 | (−0.47 to 1.47) | 0.31 | −1.01 | (−1.81 to −0.20) | 0.01 |
| Total cholesterol (mmol/L) | | | | | | | | | | | | | |
| | 1 | −0.05 | (−0.13 to 0.04) | 0.28 | −0.08 | (−0.16 to 0.00) | 0.04 | 0.02 | (−0.06 to 0.10) | 0.64 | −0.11 | (−0.19 to −0.03) | 0.009 |
| | 2 | −0.01 | (−0.10 to 0.09) | 0.90 | −0.05 | (−0.13 to 0.04) | 0.30 | 0.05 | (−0.05 to 0.14) | 0.33 | −0.07 | (−0.15 to 0.01) | 0.07 |
| Triglyceride (mmol/L)* | | | | | | | | | | | | | |
| | 1 | 0.01 | (−0.03 to 0.06) | 0.56 | −0.02 | (−0.06 to 0.03) | 0.48 | 0.05 | (0.00 to 0.09) | 0.04 | −0.05 | (−0.09 to −0.01) | 0.02 |
| | 2 | −0.01 | (−0.06 to 0.03) | 0.59 | −0.02 | (−0.06 to 0.02) | 0.38 | 0.00 | (−0.05 to 0.05) | 0.98 | −0.02 | (−0.06 to 0.02) | 0.37 |
| HDL cholesterol (mmol/L) | | | | | | | | | | | | | |
| | 1 | 0.00 | (−0.02 to 0.02) | 0.97 | 0.01 | (−0.02 to 0.03) | 0.59 | −0.01 | (−0.03 to 0.01) | 0.52 | 0.01 | (−0.01 to 0.03) | 0.23 |
| | 2 | 0.01 | (−0.01, 0.04) | 0.29 | 0.01 | (−0.02 to 0.03) | 0.56 | 0.02 | (−0.01 to 0.05) | 0.12 | 0.00 | (−0.03 to 0.02) | 0.37 |
| Fasting glucose (mmol/L) | | | | | | | | | | | | | |
| | 1 | −0.06 | (−0.12 to 0.01) | 0.08 | −0.07 | (−0.13 to −0.01) | 0.02 | −0.01 | (−0.08 to 0.05) | 0.61 | −0.07 | (−0.13 to −0.01) | 0.02 |
| | 2 | −0.04 | (−0.12 to 0.03) | 0.28 | −0.05 | (−0.12 to 0.02) | 0.16 | −0.01 | (−0.09 to 0.06) | 0.71 | −0.04 | (−0.11 to 0.02) | 0.18 |
| 120 min glucose (mmol/L) | | | | | | | | | | | | | |
| | 1 | −0.26 | (−0.42 to −0.10) | 0.002 | −0.31 | (−0.47 to −0.15) | <0.001 | −0.11 | (−0.27 to 0.06) | 0.20 | −0.27 | (−0.43 to −0.11) | 0.001 |
| | 2 | −0.26 | (−0.45 to −0.07) | 0.01 | −0.21 | (−0.39 to −0.03) | 0.02 | −0.17 | (−0.37 to 0.02) | 0.08 | −0.11 | (−0.28 to 0.05) | 0.17 |
| Fasting insulin (pmol/L)* | | | | | | | | | | | | | |
| | 1 | 0.00 | (−0.08 to 0.07) | 0.92 | −0.01 | (−0.09 to 0.06) | 0.70 | 0.01 | (−0.06 to 0.08) | 0.75 | −0.03 | (−0.10 to 0.05) | 0.47 |
| | 2 | −0.01 | (−0.09 to 0.07) | 0.82 | 0.02 | (−0.06 to 0.09) | 0.70 | −0.06 | (−0.15 to 0.02) | 0.16 | 0.05 | (−0.02 to 0.12) | 0.19 |
| HOMA-IR* | | | | | | | | | | | | | |
| | 1 | −0.01 | (−0.08 to 0.06) | 0.79 | −0.02 | (−0.09 to 0.05) | 0.61 | 0.01 | (−0.07 to 0.08) | 0.86 | −0.03 | (−0.10 to 0.04) | 0.44 |
| | 2 | −0.01 | (−0.09 to 0.07) | 0.82 | 0.02 | (−0.06 to 0.09) | 0.69 | −0.06 | (−0.14 to 0.03) | 0.17 | 0.05 | (−0.02 to 0.12) | 0.20 |
| HOMA-B | | | | | | | | | | | | | |
| | 1 | −0.11 | (−4.95 to 4.73) | 0.96 | −0.93 | (−5.76 to 3.90) | 0.71 | 1.04 | (−3.80 to 5.89) | 0.67 | −1.94 | (−6.76 to 2.89) | 0.43 |
| | 2 | −1.33 | (−7.08 to 4.43) | 0.65 | 0.34 | (−5.66 to 4.98) | 0.90 | −3.69 | (−9.57 to 2.19) | 0.22 | 1.78 | (−3.10 to 6.67) | 0.47 |
| Insulinogenic Index† | | | | | | | | | | | | | |
| | 1 | 0.04 | (−0.06 to 0.15) | 0.43 | 0.06 | (−0.04 to 0.17) | 0.25 | 0.00 | (−0.10 to 0.11) | 0.94 | 0.07 | (−0.04 to 0.17) | 0.20 |

Continued

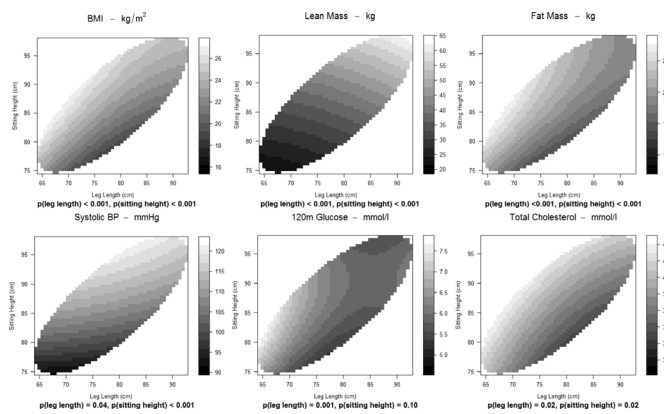

**Figure 1** Relationships of leg length and sitting height with selected cardiovascular disease risk factors. BMI, body mass index; BP, blood pressure.

associations with CVD risk factors (higher insulin resistance and lower HDL cholesterol) and some showing no associations.[11 12 14] Shorter leg length has been associated with higher low-density lipoprotein cholesterol, and greater sitting height with higher diabetes risk and dyslipidaemia in China,[37] while a study in Ghanaian adults showed greater sitting height was associated with lower CVD risk.[38] Other studies in LMICs have shown no relationships between height components and CVD risk.[5 6]

Similar to our findings, longer leg length has been associated with lower BMI, and greater sitting height has been associated with higher BMI.[22 23] No previous studies have examined associations between components of adult height and CVD risk in relation to direct measurements of fat and lean mass. While longer leg length was associated with greater lean mass and lower body fat per cent, this did not appear to explain the inverse association between leg length and CVD risk factors—only the relationship between leg length and Matsuda Index was attenuated after adjustment for lean and fat mass. In contrast, our data indicate that the relationship between sitting height and CVD risk factors could be partly explained by higher fat mass.

Secular increases in height in high-income countries are now slowing down.[39] Much of this increase is reflected in greater leg length[7]; this increment has been shown to occur by 2 years of age.[7 40 41] Leg length increases faster than sitting height in infancy, suggesting that it may be related to better fetal and/or infant health and nutrition.[42] In contrast, sitting height increases more than leg length in later childhood[40–42] due to faster growth of vertebrae than long bones during the pubertal growth spurt.[19 22 40 41] However, direct evidence that leg length and sitting height reflect conditions in early life and later childhood, respectively, is lacking. Birth weight, which might be expected to predict longer leg length, is equally strongly predictive of sitting height,[12 15] as in our study. The independent relationships of leg length and sitting height with CVD risk factors suggest that patterns and determinants of linear growth are important and may also influence body composition and CVD risk.

**Table 3** Continued

| Outcome | Model | Height (z) B | 95% CI | P | Leg length (z) B | 95% CI | P | Sitting height (z) B | 95% CI | P | Leg length to sitting height (z) B | 95% CI | P |
|---|---|---|---|---|---|---|---|---|---|---|---|---|---|
| | 2 | 0.08 | (−0.05 to 0.21) | 0.24 | 0.10 | (−0.02 to 0.22) | 0.12 | 0.00 | (−0.14 to 0.13) | 0.95 | 0.09 | (−0.02 to 0.20) | 0.11 |
| Matsuda Index* | 1 | **0.08** | **(0.01 to 0.16)** | **0.04** | **0.11** | **(0.03 to 0.18)** | **0.005** | 0.02 | (−0.06 to 0.09) | 0.62 | **0.11** | **(0.03 to 0.18)** | **0.005** |
| | 2 | 0.05 | (−0.04 to 0.13) | 0.27 | 0.03 | (−0.04 to 0.11) | 0.39 | 0.04 | (−0.04 to 0.13) | 0.32 | 0.01 | (−0.06 to 0.08) | 0.80 |
| Disposition Index | 1 | 0.01 | (−0.58 to 0.60) | 0.96 | 0.16 | (−0.43 to 0.74) | 0.61 | −0.18 | (−0.77 to 0.41) | 0.55 | 0.32 | (−0.27 to 0.90) | 0.29 |
| | 2 | 0.04 | (−0.06 to 0.14) | 0.41 | 0.03 | (−0.06 to 0.12) | 0.55 | 0.03 | (−0.07 to 0.12) | 0.60 | 0.00 | (−0.08 to 0.08) | 0.93 |

Components of height were z standardised. Model 1 was adjusted for age, sex and interaction of age and sex. Model 2 was adjusted for age, sex, interaction of age and sex, lean mass, fat mass, interaction of sex and lean mass, and interaction of sex and fat mass.

*Outcome variables were log transformed to ensure normality.

†Outcome variable was Fisher-Yates transformed to ensure normality.

BP, blood pressure; HDL, high-density lipoprotein; HOMA-B, homeostatic model assessment of β-cell function; HOMA-IR, homeostatic model assessment of insulin resistance.

**Table 4** Relationship of parental height and its components and intergenerational change in height with offspring cardiovascular disease parameters at 21 years

| Outcome at 21 years | Mid-parental height | Father's height | Father's leg length | Father's sitting height | Mother's height | Mother's leg length | Mother's sitting height | Intergenerational change in height | Intergenerational change in leg length | Intergenerational change in sitting height |
|---|---|---|---|---|---|---|---|---|---|---|
| BMI | 0.07 | 0.03 | −0.03 | 0.09 | 0.00 | −0.04 | 0.05 | −0.02 | −0.13* | 0.16** |
| SLI score | 0.15** | 0.13* | 0.04 | 0.18** | 0.10 | 0.07 | 0.10 | 0.00 | 0.05 | −0.04 |
| Systolic BP | 0.07 | 0.04 | −0.05 | 0.13* | 0.06 | 0.01 | 0.10 | −0.00 | −0.09 | 0.03 |
| Diastolic BP | 0.02 | 0.02 | −0.02 | 0.08 | −0.04 | −0.09 | 0.05 | 0.05 | −0.09 | 0.03 |
| Cholesterol | 0.01 | −0.03 | −0.11 | 0.08 | 0.02 | 0.002 | 0.04 | −0.06 | −0.07 | −0.01 |
| Triglycerides | 0.08 | 0.01 | −0.01 | 0.04 | 0.04 | −0.00 | 0.08 | 0.02 | −0.03 | 0.06 |
| HDL cholesterol | −0.01 | −0.02 | −0.04 | 0.02 | 0.05 | 0.06 | 0.02 | −0.01 | 0.02 | −0.02 |
| Fasting glucose | −0.04 | −0.10 | −0.08 | −0.09 | −0.00 | −0.03 | 0.03 | −0.06 | −0.09 | 0.01 |
| 120 min glucose | −0.07 | −0.10 | −0.13* | −0.03 | −0.05 | −0.07 | −0.01 | −0.09 | −0.09 | −0.02 |
| Fasting insulin | 0.00 | 0.01 | 0.04 | −0.03 | 0.01 | −0.02 | 0.03 | −0.09 | −0.12* | −0.00 |
| 120 min insulin | 0.02 | −0.04 | −0.05 | −0.01 | 0.03 | −0.04 | 0.09 | −0.03 | −0.05 | 0.02 |
| HOMA-IR | 0.00 | 0.00 | 0.04 | −0.04 | 0.01 | −0.02 | 0.04 | −0.10 | −0.13* | −0.00 |
| HOMA-B | 0.02 | 0.03 | 0.05 | 0.00 | 0.01 | 0.01 | −0.01 | −0.04 | −0.06 | 0.04 |
| Insulinogenic Index | 0.11 | 0.11 | 0.13 | 0.03 | 0.05 | 0.05 | 0.03 | −0.10 | −0.03 | −0.03 |
| Matsuda Index | 0.04 | 0.10 | 0.07 | 0.09 | −0.01 | 0.01 | −0.02 | 0.05 | 0.08 | −0.05 |
| Disposition Index | 0.06 | 0.10 | 0.10 | 0.07 | 0.02 | 0.02 | 0.02 | −0.07 | 0.00 | −0.10 |

Values are Pearson correlation coefficients adjusted for age and gender, as appropriate; exposures and outcomes are Z standardised.
*P<0.05, **P<0.01, ***P<0.001.
BP, blood pressure; HDL, high-density lipoprotein; HOMA-B, homeostatic model assessment of β-cell function; HOMA-IR, homeostatic model assessment of insulin resistance; SLI, Standard of Living Index.

Our data do not elucidate the mechanisms linking greater sitting height to increased adiposity or greater leg length to lower CVD risk. Lower birth and infant weight are associated with lower adult lean mass, while greater weight gain in later childhood is associated with increased adult adiposity in several countries, including LMICs.[43] Both are associated with a higher prevalence of CVD, type 2 diabetes and their risk factors.[44 45] These findings led to the developmental programming hypothesis that early undernutrition permanently alters the body's structure and metabolism, leaving an increased vulnerability to the adverse effects of 'excess' nutrition in later life.[46] This sequence has been suggested as a possible explanation for the rising epidemics of CVD and diabetes, and high risk factor levels (as exemplified by our cohort), in LMICs. If it is true that lower leg length reflects poorer early life environment, and greater sitting height reflects accelerated puberty and increased adiposity, our findings linking CVD risk factors to these height components would be broadly consistent with the developmental programming hypothesis. Our findings do not suggest that the relationship between short height and CVD risk reflects lower socioeconomic status. First, we found no relationships of socioeconomic status with CVD risk factors. Second, we would expect both short leg length and short sitting height to be associated with higher CVD risk. Although genetic explanations for the associations between components of height and CVD risk factors are possible, common genes that influence both height and CVD explain little of the effect.[47] We have not explored genetic determinants in our study. Given the age of our cohort, reverse causality is unlikely to be a factor.

A greater increase in leg length between generations was associated with lower fasting insulin and insulin resistance. No previous studies have examined changes in leg length or sitting height between generations. A study in Scotland showed that although short height of both parents was independently associated with increased risk of coronary heart disease in the offspring, the association was stronger with mothers' height.[48] A study from India showed that shorter maternal but not paternal height was associated with higher risk of glucose intolerance in the offspring.[49] A greater increase in leg length between generations may reflect better fetal and infant growth, and this may have persisting effects on offspring development and health. This may also be reflected through greater insulin sensitivity promoting greater growth.

The PCS is one of few cohorts in LMICs with follow-up from childhood to adult life and measurements of adult height components and parental heights. Our loss to follow-up of ~25% is low for a long-term longitudinal study and adds to internal validity. Our attrition was mainly due to loss to follow-up in urban Pune with very few refusals. Moreover, our study sample was similar in key characteristics to those who were lost to follow-up. Limitations were a lack of data on length and height components at birth and longitudinally throughout the growing period. While we have direct measurements of lean and fat mass, we are unable to differentiate between visceral and parietal fat components. The participants were all born in one hospital in Pune and were available for follow-up from childhood to young adulthood; this limits representativeness, given the population size and socioeconomic diversity of India. However, the KEM is the second largest hospital in Pune and offers services to a wide range of socioeconomic classes.

## CONCLUSIONS

Ours is the first study to examine associations of CVD risk factors with components of adult height, using body composition measures to understand the relationship. Shorter leg length and greater sitting height were associated with a more adverse CVD risk factor profile. We speculate that the associations with leg length and sitting height reflect the benefits of better growth in early childhood, which is protective, and accelerated growth in later childhood which may be detrimental, and that these relationships are partially explained by body composition. Perhaps the strongest message that emerges from this analysis is the need to gain a better understanding of the determinants of height components.

**Author affiliations**
[1]MRC Lifecourse Epidemiology Unit, University of Southampton, Southampton, Hampshire, UK
[2]Epidemiology Research Unit, CSI Holdsworth Memorial Hospital, Mysore, Karnataka, India
[3]Diabetes Unit, KEM Hospital Research Centre, Pune, Maharashtra, India
[4]Department of Biostatistics, Vanderbilt University, Nashville, Tennessee, USA
[5]Vadu Rural Health Centre, KEM Hospital Research Centre, Pune, Maharashtra, India
[6]Department of Statistics, BKL Walawalkar Hospital and Diagnostic Centre, Ratnagiri, Maharashtra, India
[7]Department of Radiodiagnosis, Bharati Medical College and Hospital, Bharati Vidyapeeth, Pune, Maharashtra, India
[8]Department of Paediatrics, KEM Hospital, Pune, Maharashtra, India

**Acknowledgements** We are grateful to the study participants for taking part in this study. We thank Dr KJ Coyaji, Medical Director of the KEM Hospital, and Dr VS Padbidri, Director Research for providing research facilities. We thank Mrs PC Yajnik, Ms LV Ramdas, Ms Sonali Hardikar, Ms Sonali Wagle, Ms.Vaishali Deshpande, Ms Nilam Memane, Ms Deepa Raut, Ms Pallavi Hardikar, Ms Sonal Kasture, Ms Vaishali Kantikar, Ms Komal Advani, Ms Neha Gurav, Ms Rupali Saswade and Mr Arjunsing Patilkendre for their contribution in the collection of the data. We also thank Mr TM Deokar, Mr SD Chougule, Mr AB Gaikwad, Mr ML Hoge, Mr. Vipul Wagh, Mr SN Khemkar, Mr SB Wagh, Mr BS Jadhav and Ms Prachi Katre for their invaluable contribution to the study. Dr Sharad Gore helped with statistical queries. We also acknowledge the support of Sneha-India.

**Contributors** CY and CF designed the study, analysed and interpreted the data and wrote the manuscript. KK, SMJ and CDG analysed and interpreted the data and wrote the manuscript. HL, DB, CJ and AK carried out data collection and critically reviewed the manuscript. CO analysed and interpreted the data and critically reviewed the manuscript. AB, SB and AP designed the study and critically reviewed the manuscript. CY is the guarantor of this work and as such, had full access to all the data in the study and takes responsibility of the integrity of the data and the accuracy of the data analysis. All authors approved the final version.

**Funding** The study was funded by The Wellcome Trust, UK (grant number 083460/Z/07/Z), the Medical Research Council, UK, and the Department for International Development, UK.

**Competing interests** None declared.

**Patient and public involvement** Patients and/or the public were not involved in the design, conduct, reporting or dissemination plans of this research.

**Patient consent for publication** Not required.

**Ethics approval** Ethical permission was obtained from the KEM Hospital Ethics Committee and informed consent was obtained from all participants.

**Provenance and peer review** Not commissioned; externally peer reviewed.

**Data availability statement** Data are available upon reasonable request. The cohort is a hospital-based birth cohort with data collected at 4, 8 and 21 years on anthropometry, dietary information and cardiometabolic risk factors. Data collected earlier at birth and at 4 and 8 years have been published, and we are in the process of analysing and publishing data collected at 21 years. For access to data, please contact Prof CS Yajnik, Director, Diabetes Unit, KEM Hospital Research Centre, Pune, India. E-mail: diabetes@kemdiabetes.org; csyajnik@gmail.com.

**ORCID iDs**
Kalyanaraman Kumaran http://orcid.org/0000-0002-6107-8608
Chittaranjan Yajnik http://orcid.org/0000-0002-2911-2378

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
