## [Reviewer comments · BMJ Open]

ARTICLE DETAILS

TITLE (PROVISIONAL)	DO COMPONENTS OF ADULT HEIGHT PREDICT BODY COMPOSITION AND CARDIOMETABOLIC RISK IN A YOUNG ADULT SOUTH ASIAN INDIAN POPULATION? FINDINGS FROM A HOSPITAL-BASED COHORT STUDY IN PUNE, INDIA: THE PUNE CHILDREN'S STUDY.
AUTHORS	Kumaran, Kalyanaraman; Joshi, Suyog; Di Gravio, Chiara; Lubree, Himangi; Joglekar, Charudatta; Bhat, Dattatray; Kinare, Arun; Bavdekar, Ashish; Bhave, Sheila; Pandit, Anand; Osmond, Clive; Yajnik, Chittaranjan; Fall, Caroline

VERSION 1 – REVIEW

REVIEWER	Brie Reid University of Minnesota, USA
REVIEW RETURNED	26-Mar-2020

GENERAL COMMENTS	1. Is the research question or study objective clearly defined? This paper does not name a focal exposure measure and tests many exposures. There is a lack of clarity around hypothesis testing. This paper would benefit from a hypothesis section – which exposures are hypothesized to be associated with which outcomes? The dataset seems to be a rich one, and the aims of the paper (developmental programming, intergenerational risk of CVD). A sentence comparing why specific aspects of adult stature (leg length vs. sitting height) may contribute to CVD or why they are different would be appreciated in the introduction. 3. Is the study design appropriate to answer the research question? The paper introduction discusses early life programming of CVD, but all of the exposure measures are conducted concurrently with CVD risk. Why not either (a) use data from 8y to predict CVD risk and/or (b) test a mediation hypothesis with data from childhood predicting CVD risk, mediated through leg length? The study design describes itself as a prospective cohort study, but the key statistical model to test the hypothesis uses essentially cross-sectional data, and does not take advantage of the prospectively-collected data. 4. Are the methods described sufficiently to allow the study to be repeated?
---

	A few methods questions remained:  - How long were participants asked to fast for? - Were z-scores calculated based on national norms or within the sample? Please clarify in the methods section - Where was socioeconomic status used in the statistical models? - Where there exclusionary characteristics of participants? I.e. were participants excluded for any reason (pregnancy, disease, etc.)? - What were the reasons for the attrition of participants? Simple loss to follow-up? Out-migration? Refusal of study? - In Table 1, what does CDC Z score refer to? Z score of height in childhood? Height in adulthood? Reasons for use of the CDC z-scores would be helpful. 6. Are the outcomes clearly defined? The paper does not define either a primary or a secondary outcome measure and instead lists a number of outcomes. 7. If statistics are used are they appropriate and described fully? What is the rationale for running multiple models with multiple exposures? Greater justification of why the exposure is being used would be helpful. It is unclear why the authors separate the analysis of the prospective data collected in childhood and the CVD risk data. 10. Are they presented clearly? The results section would read clearer if the language “directly related” was changed to positively associated and “inversely related” was changed to “negative associated” to clarify the statistical direction of the association. What does “markedly attenuated” by adjustment mean? Is it completely attenuated? Or partially? Please number the models (e.g. model 1, model 2, model 3) in the statistical methods section when discussion the models fit for ease of reading the table in the results section 12. Are the study limitations discussed adequately? The authors speculate that early childhood growth plays a role in leg length and CVD risk but did not test this systematically in a model. This limitation was not discussed.
--	--

REVIEWER	Shinya Suzuki The Cardiovascular Institute, Japan
REVIEW RETURNED	28-Mar-2020

GENERAL COMMENTS	This study investigated the association between components of adult height and CVD risk factors. Based on the results, they speculated that the associations with leg length and sitting height reflect the benefits of better growth in early childhood which is protective, and accelerated growth in later childhood which may be detrimental, and that these relationships are partially explained by body composition.
---

	Although the data are many and complex, the authors well summarized the results. The conclusions are sound and reasonable. 1) In the discussion, the authors repeated the words of "associated" or "related". But the size of the association seems to be sometimes small and sometimes not small. It might be better if the authors can use the terms of quantitative evaluation. 2) In Table 3, the authors analyzed the data combining men and women. I think the association between height and CVD risk would be strongly affected by sex category, and I am afraid it cannot be "adjusted" by multivariable model if the effect is very strong. The authors should carefully explain for this point.
--	---

REVIEWER	Yasuo Okumura Nihon University School of Medicine
REVIEW RETURNED	30-Mar-2020

GENERAL COMMENTS	Comments to the Author Kumaran et al. reported a prospective cohort study, "the Pune Children's Study (PCS)", regarding the relationship between components of height and cardiovascular disease (CVD) risks. They described that both a shorter adult leg length and greater sitting height are associated with a more adverse CVD risk factors profile. Regarding the relationship between sitting height/leg length and CVD risk factors, there are several other studies shown as the references of this paper. They emphasize this study is one of the few cohorts from low- and middle-income countries, but a comparison of regional differences in sitting height/leg length and CVD risk factor has been reported in previous other report (Boateng D, et al. Doi: 10.1017 / S2040174419000527.). This study is limited to its value as a local epidemiological study, but the study may help to understand the relationship between the body type and CVD risk in India. As discussed, they have not shown the exact mechanisms of leg length and sitting height affect CVD risk factors. They have discussed that, according to the other previous reports (reference 15 · 16 in their paper), a longer leg length was associated with lower BMI, and greater sitting height with higher BMI. Even if the environment at the young age affects the sitting height/leg length or CVD risk factor, we generally manage the CVD risk factors by using BMI as an index, rather than by sitting height and leg length. Subsequent data, blood pressure, glycemic, and lipid abnormalities might have only reflected the relationship between obesity and CVD risk factors, as previously shown by BMI. As an improvement, in order to use the epidemiological findings of this study as indicators of CVD risk, they need to clarify what is the sitting height and leg length reflect body composition. It is a known fact that abdominal circumference and visceral fat affect CVD risk factors. Visceral fat, which is related to obesity, is mainly accumulated in the abdominal trunk. In general, upper and lower limbs consist lower fat mass than in the abdominal trunk. It would be better to measure lean and fat mass separately for the abdominal trunk, upper limbs, and lower limbs by DXA. Comparing the amount and ratio of fat mass and lean mass by portion, the relationship between sitting height/leg length and fat mass may
---

	become clearly. It may provide mechanistic insights into understanding the relationship between fat/lean mass and CVD risk factor. Major Comments  1. It is necessary to clarify the relationship between sitting height/leg length and fat/lean mass. Data on fat mass and lean mass in DXA must be presented separately for the abdominal trunk, upper limbs, and lower limbs. The relationship between fat mass and CVD risk factors needs to be clearly considered. 2. It is necessary to present the similarities/differences of the components of height and CVD risk factors between this study and the data from high-income countries. 3. They should highlight a limitation that their data may not be applicable to other countries or regions in India because this study is derived from a single center analysis. It would be better to show their limitations as a separate section. Minor Comments  1. I think the races background of the subjects of this study should be clear. The data of this study is based on the people born in one hospital in India, and it is presumed that they are all Indians.
--	--

VERSION 1 – AUTHOR RESPONSE

Reviewer 1

1. Is the research question or study objective clearly defined?

This paper does not name a focal exposure measure and tests many exposures. There is a lack of clarity around hypothesis testing. This paper would benefit from a hypothesis section – which exposures are hypothesized to be associated with which outcomes?

Response: We apologise for the lack of clarity. Our main exposures are components of height – leg length and sitting height. Our outcomes are cardiometabolic risk factors – blood pressure, glycaemia and lipid profile. We hypothesise:

1. Longer leg length will be associated with lower, and greater sitting height will be associated with higher cardiometabolic risk factors.
2. The relationship between leg length and sitting height (components of height) and cardiometabolic risk factors may be influenced by body composition i.e. lean and fat mass.

We have now explicitly mentioned our hypotheses in the introduction (page 5).

The dataset seems to be a rich one, and the aims of the paper (developmental programming, intergenerational risk of CVD). A sentence comparing why specific aspects of adult stature (leg length vs. sitting height) may contribute to CVD or why they are different would be appreciated in the introduction.

Response: The reasons for the possible associations are not clear and remain speculative. The developmental origins of health and disease hypothesis suggests that during intrauterine insults, blood flow to the brain is selectively preserved at the expense of the peripheral organs (brain sparing effect). This results in shorter leg length at birth and may be one reason for the possible association between shorter leg length and higher cardiometabolic risk factor levels. It is also suggested that much of the growth in leg length occurs during early life while the increase in sitting height occurs during the pubertal growth spurt. We have now added a couple of sentences to the introduction section as suggested by the reviewer (page 4).

2. Is the study design appropriate to answer the research question?

The paper introduction discusses early life programming of CVD, but all of the exposure measures are conducted concurrently with CVD risk. Why not either (a) use data from 8y to predict CVD risk and/or (b) test a mediation hypothesis with data from childhood predicting CVD risk, mediated through leg length?

Response: In this paper, we set out to explore the relationship between components of height in young adulthood (final height) with cardiometabolic risk factors, and examine the possible influence of lean and fat mass. We have examined the relationships between size at birth, and change in linear growth, lean mass and fat mass between birth and 8 years, and between 8 and 21 years on cardiometabolic risk factors in a separate paper which has been accepted by the Journal of Developmental Origins of Health and Disease (in print). We regret we do not have data on leg length or sitting height at 8 years.

The study design describes itself as a prospective cohort study, but the key statistical model to test the hypothesis uses essentially cross-sectional data, and does not take advantage of the prospectively-collected data.

Response: We agree that we did not use the prospective data in this paper. However the main purpose in this paper was to examine the relationship between components of final height and cardiometabolic risk factors and the possible influence of direct measurements of body composition on these relationships. We have used the prospective data in a separate manuscript as mentioned above. We have clarified the design in the abstract.

3. Are the methods described sufficiently to allow the study to be repeated?

A few methods questions remained:

- How long were participants asked to fast for?

Response: All participants fasted for ~10 hours. They were brought into the research unit the previous night and given a standard meal (around 9 pm). Fasting blood samples were drawn the following morning (7-8 am). We have now clarified this in the methods section (page 5).

- Were z-scores calculated based on national norms or within the sample? Please clarify in the methods section

Response: The z-scores were based on calculations within the sample. We have now clarified this in the methods section (page 7).

- Where was socioeconomic status used in the statistical models?

Response: The models were adjusted for SLI (Standard of Living Index, the measure of socio-economic status used in the study) scores. This has previously been mentioned in the results section (page 12).

- Where there exclusionary characteristics of participants? I.e. were participants excluded for any reason (pregnancy, disease, etc.)?

Response: Those who were pregnant were not examined at that time, but were called for investigations 6 months after delivery. We have now clarified this in the methods section (page 5). There were no other exclusions.

- What were the reasons for the attrition of participants? Simple loss to follow-up? Out-migration? Refusal of study?

Response: Attrition was primarily due to loss of follow-up in urban Pune. There were very few refusals. We have now added a sentence in the discussion (page 19).

- In Table 1, what does CDC Z score refer to? Z score of height in childhood? Height in adulthood? Reasons for use of the CDC z-scores would be helpful.

Response: The CDC z-score refers to height at 20 years. We used the CDC reference for comparison as our participants were ~21 years (final adult height reached) and the WHO reference stops at 18 years. We have now clarified this in the methods section (page 7).

4. Are the outcomes clearly defined?

The paper does not define either a primary or a secondary outcome measure and instead lists a number of outcomes.

Response: Our intention was to examine cardiometabolic risk factors – blood pressure, glycaemia and lipid profile. We accept that the number of models and outcomes make Table 3 look confusing. We have now simplified Table 3 to depict two models: 1) Model 1, examining relationships of height components with cardiometabolic risk factors adjusted for age, sex and age-sex interaction; 2) Model 2 additionally adjusting for lean and fat mass, and their interaction with sex.

5. If statistics are used are they appropriate and described fully?

What is the rationale for running multiple models with multiple exposures? Greater justification of why the exposure is being used would be helpful.

Response: We used the multiple models to explore relationships of components of height with cardiometabolic risk factors in a step-wise manner adding potential confounding and influencing factors. In line with the reviewer's comment, we have now simplified the models as mentioned above.

It is unclear why the authors separate the analysis of the prospective data collected in childhood and the CVD risk data.

Response: Relationships between birth size, growth in childhood and young adulthood and cardiometabolic risk factors using the prospective data have been examined in a separate manuscript (accepted in *J DOHAD*). The current analysis aimed to explicitly explore the relationships between components of adult height and cardiometabolic risk factors, and to examine the possible influence of lean and fat mass on these relationships. To our knowledge, this has not been examined previously.

6. Are they presented clearly?

The results section would read clearer if the language “directly related” was changed to positively associated and “inversely related” was changed to “negative associated” to clarify the statistical direction of the association.

Response: We thank the reviewer for this helpful suggestions and have amended the language in the manuscript accordingly.

What does “markedly attenuated” by adjustment mean? Is it completely attenuated? Or partially?

Response: We used ‘markedly attenuated’ to indicate that the relationship was significantly attenuated but not completely. We have now used ‘significantly’ instead of markedly. Any other suggestion would be welcome.

Please number the models (e.g. model 1, model 2, model 3) in the statistical methods section when discussion the models fit for ease of reading the table in the results section

Response: We thank the reviewer for this suggestions and have amended the statistical methods section accordingly (page 7). As mentioned previously, we have now simplified Table 3 and reduced the number of models.

7. Are the study limitations discussed adequately?

The authors speculate that early childhood growth plays a role in leg length and CVD risk but did not test this systematically in a model. This limitation was not discussed.

Response: Unfortunately we do not have data on leg length and sitting height at 8 years. We acknowledge this limitation and have previously mentioned it in the discussion section under limitations (page 20).

Reviewer 2

Based on the results, they speculated that the associations with leg length and sitting height reflect the benefits of better growth in early childhood which is protective, and accelerated growth in later childhood which may be detrimental, and that these relationships are partially explained by body composition.

Although the data are many and complex, the authors well summarized the results. The conclusions are sound and reasonable.

Response: We thank the reviewer for their positive comments.

1) In the discussion, the authors repeated the words of "associated" or "related". But the size of the association seems to be sometimes small and sometimes not small. It might be better if the authors can use the terms of quantitative evaluation. Modify where appropriate.

Response: We thank the reviewer for this suggestion, and have made amendments using 'significantly' where appropriate. The modified table 3 shows the size of the effects with 95% CIs.

2) In Table 3, the authors analyzed the data combining men and women. I think the association between height and CVD risk would be strongly affected by sex category, and I am afraid it cannot be "adjusted" by multivariable model if the effect is very strong. The authors should carefully explain for this point.

Response: Initially, our analyses were separated by sex. The directions of the results in both sexes were similar. Moreover, we also tested for interaction with sex and found no significant interactions. We had mentioned previously that the results were in similar directions in both sexes (page 11); we have now added a sentence to say that there were no interactions with sex (page 11).

Reviewer 3

Kumaran et al. reported a prospective cohort study, "the Pune Children's Study (PCS)", regarding the relationship between components of height and cardiovascular disease (CVD) risks. They described that both a shorter adult leg length and greater sitting height are associated with a more adverse CVD risk factors profile. Regarding the relationship between sitting height/leg length and CVD risk factors, there are several other studies shown as the references of this paper. They emphasize this study is one of the few cohorts from low- and middle-income countries, but a comparison of regional differences in sitting height/leg length and CVD risk factor has been reported in previous other report (Boateng D, et al. Doi: 10.1017 / S2040174419000527.). This study is limited to its value as a local epidemiological study, but the study may help to understand the relationship between the body type and CVD risk in India.

As discussed, they have not shown the exact mechanisms of leg length and sitting height affect CVD risk factors. They have discussed that, according to the other previous reports (reference 15 · 16 in their paper), a longer leg length was associated with lower BMI, and greater sitting height with higher BMI. Even if the environment at the young age affects the sitting height/leg length or CVD risk factor, we generally manage the CVD risk factors by using BMI as an index, rather than by sitting height and leg length. Subsequent data, blood pressure, glycemic, and lipid abnormalities might have only reflected the relationship between obesity and CVD risk factors, as previously shown by BMI.

As an improvement, in order to use the epidemiological findings of this study as indicators of CVD risk, they need to clarify what is the sitting height and leg length reflect body composition. It is a known fact that abdominal circumference and visceral fat affect CVD risk factors. Visceral fat, which is related to obesity, is mainly accumulated in the abdominal trunk. In general, upper and lower limbs consist lower fat mass than in the abdominal trunk. It would be better to measure lean and fat mass

separately for the abdominal trunk, upper limbs, and lower limbs by DXA. Comparing the amount and ratio of fat mass and lean mass by portion, the relationship between sitting height/leg length and fat mass may become clearly. It may provide mechanistic insights into understanding the relationship between fat/lean mass and CVD risk factor.

Response: We thank the reviewer for their positive comments. We agree that direct measurements of lean and fat mass reflect body composition better than BMI alone. We have responded to the suggestion about regional lean and fat mass in more detail below.

The paper Boateng D, et al. Doi: 10.1017 / S2040174419000527 examined relationships of height components and risk of CVD in Ghanaian adults in Europe and Ghana. However, this study did not have direct measurements of body composition. We have now added this reference to the discussion section.

Major Comments

1. It is necessary to clarify the relationship between sitting height/leg length and fat/lean mass. Data on fat mass and lean mass in DXA must be presented separately for the abdominal trunk, upper limbs, and lower limbs. The relationship between fat mass and CVD risk factors needs to be clearly considered.

Response: In our earlier analyses, we examined the relationship between height components and cardiometabolic risk factors adjusting for total fat and lean mass. We also considered fat mass percent as an index of the relative proportions of fat and lean mass. Additionally, we specifically examined the relationship between leg length and cardiometabolic risk factors adjusting for leg lean and fat mass, and between sitting height and cardiometabolic risk factors adjusting for trunk lean and fat mass. We found that adjusting for these regional measurements did not explain the relationship any more than total fat and lean mass and we had previously mentioned this finding in the results (page 12).

In line with the reviewer's comments, we have now undertaken additional analyses. We examined the relationship of height components with regional body composition measurements (arm lean and fat mass, leg lean and fat mass, and trunk lean and fat mass); the findings are similar to those seen with total lean and fat mass. We present this as a correlation matrix in the supplementary material (Supplementary Figure 1). We then adjusted for all the regional body composition measurements instead of total lean and fat mass while examining the relationship between height components and cardiometabolic risk factors. We found no specific patterns with regional body composition measurements. These analyses did not explain the relationships between height and cardiometabolic risk factors any more than total lean and fat mass. We have now added these findings in the text of the results section. However, we acknowledge that we are unable to differentiate between visceral and parietal fat distribution and this factor may be a potential limitation. We have now added a sentence to this effect in the limitations of the study (page 20).

2. It is necessary to present the similarities/differences of the components of height and CVD risk factors between this study and the data from high-income countries.

Response: We have presented the results of our study in relation to findings from other high-income countries in the discussion section (page 17).

3. They should highlight a limitation that their data may not be applicable to other countries or regions in India because this study is derived from a single center analysis. It would be better to show their limitations as a separate section.

Response: We have now modified the limitations section to incorporate the above suggestion (page 20).

Minor Comments

1. I think the races background of the subjects of this study should be clear. The data of this study is based on the people born in one hospital in India, and it is presumed that they are all Indians.

Response: Yes, all the participants are South Asian Indians; we have now clarified this in the title.

VERSION 2 – REVIEW

REVIEWER	Brie Reid University of Minnesota, USA
REVIEW RETURNED	29-Jul-2020

GENERAL COMMENTS	The authors have responded sufficiently to reviewer comments.
---

REVIEWER	Yasuo Okumura Division of Cardiology, Department of Medicine, Nihon University School of Medicine, Tokyo, Japan
REVIEW RETURNED	03-Aug-2020

GENERAL COMMENTS	There were no further comments.
---------------------------------